# Identification of Specific IgE Antibodies and Asthma Control Interaction and Association Using Cluster Analysis in a Bulgarian Asthmatic Children Cohort

**DOI:** 10.3390/antib9030031

**Published:** 2020-07-06

**Authors:** Snezhina Lazova, Tsvetelina Velikova, Stamatios Priftis, Guergana Petrova

**Affiliations:** 1Pediatric Department, UMHATEM “N. I. Pirogov”, 21 Totleben blvd, 1606 Sofia, Bulgaria; 2Sofia University—Medical Faculty, University Hospital Lozenets, 1 Kozyak str, 1407 Sofia, Bulgaria; tsvelikova@medfac.mu-sofia.bg; 3Faculty of Public Health, Medical University of Sofia, Health Technology Assessment Department, 8 Bialo more str., 1527 Sofia, Bulgaria; stamatios.priftis@gmail.com; 4Medical University, Pediatric clinic, UMHAT Alexandrovska, 1 Georgi Sofiyski blvd., 1431 Sofia, Bulgaria; gal_ps@yahoo.co.uk

**Keywords:** asthma phenotype in childhood, total IgE, cluster analysis, spirometry, asthma control

## Abstract

(1) Background: Asthma is a complex heterogeneous disease that likely comprises several distinct disease phenotypes, where the clustering approach has been used to classify the heterogeneous asthma population into distinct phenotypes; (2) Methods: For a period of 1 year, we evaluated medical history data of 71 children with asthma aged 3 to 17 years, performing pulmonary function tests, drew blood for IgE antibodies against inhalation and food allergies detection, and Asthma Control Questionnaire (ACQ); (3) Results: Five distinct phenotypes were determined. Cluster 1 (n = 10): (non-atopic) the lowest IgE level, very low ACQ, and median age of diagnosis. Cluster 2 (n = 28): (mixed) the highest Body mass index (BMI) with the latest age of diagnosis and high ACQ and bronchodilator response (BDR) levels and median and IgE levels. Cluster 3 (n = 19) (atopic) early diagnosis, highest BDR, highest ACQ score, highest total, and high specific IgE levels among the clusters. Cluster 4 (n = 9): (atopic) the highest specific IgE result, relatively high BMI, and IgE with median ACQ score among clusters. Cluster 5 (n = 5): (non-atopic) the earliest age for diagnosis, with the lowest BMI, the lowest ACQ score, and specific IgE levels, with high BDR and the median level of IgE among clusters; (4) Conclusions: We identified asthma phenotypes in Bulgarian children according to IgE levels, ACQ score, BDR, and age of diagnosis.

## 1. Introduction

Asthma is a heterogeneous disease that can be classified into several different phenotypes according to the disease’s clinical spectrum, inflammatory class, demographic characteristics, and comorbidity presence [1]. By definition, a phenotype is a complex of observable characteristics in an organism that results from the interaction of genotype and environmental factors [2]. In line with this, asthma phenotypes can be the result of many factors such as age at onset of asthma symptoms, atopy, type of inflammation found in the airways, disease severity, standard therapy response, etc. It is expected that in the future, phenotype categorization could be used as a means of disease prevention, control, and treatment selection.

Due to the variable clinical presentation of childhood asthma, there is a growing scientific and clinical interest in trying to uncover new asthma phenotypes and endotypes to target therapy individually [2,3]. Most of the asthma phenotypes descriptions so far are based on the time when “wheezing” occurs during childhood [4]. Other phenotypes are based on the presence or absence of allergic sensitization, eosinophilic, or non-eosinophilic inflammation, response to therapy, the severity of asthma, and other allergic comorbidities [5,6,7,8]. Despite the accumulated data and experience, there are still serious difficulties in determining optimal therapy, monitoring strategy and asthma follow-up. Especially challenging is the situation in young children with asthma-like symptoms, for in the first few years of life, wheezing disorders are very common [9]. In childhood, reversible bronchial obstruction may be a common clinical manifestation of various diseases, with different etiologies and different genetic backgrounds. Thus, it is so essential to establish asthma diagnosis early.

In recent years, approaches to identifying asthma subtypes have evolved from the adoption of subjective expertise to the application of more objective methodologies such as machine learning driven by specific data and research results [10,11]. Statistical methods based on machine learning facilitate efficient extraction of data for identification and analysis of different disease models. These include principal component analysis, exploratory factor analysis (EFA), hierarchical clustering (HC), etc. The process of splitting a given set of physical or abstract objects into groups of similar objects is called clustering. A cluster is a collection of objects that look alike inside a cluster and are different from objects in other clusters. Cluster analysis uses a multivariable mathematical algorithm that is generally applied to two main functions: determining qualitative similarity between subjects in the study population based on multiple specific variables and grouping subjects into clusters according to their general characteristics [12,13,14].

In cluster analysis, the goal is to group objects into groupscalled clusters, using a number of features. These characteristics, called variables, are age, gender, BMI, baseline spirometry parameters, bronchodilator response (BDR), biomarkers that reflect eosinophilic inflammation, atopic status, asthma control level, etc. BDR reflects many of these biomarkers like eNO, sputum and bronchial eosinophils. Two major studies using factor and multifactorial analysis have demonstrated the importance of BDR in asthma phenotyping [12]. Determining atopic status increases the likelihood of having asthma in patients with respiratory symptoms. Total IgE amount and allergen-specific IgE antibodies presence in serum are relevant biomarkers for defining the phenotype of patients with asthma symptoms [14].

We aimed to perform cluster analysis to identify subtypes of childhood asthma in the study population, based on crucial disease characteristics such as symptoms (control level, Asthma Control Questionnaire (ACQ) score), pulmonary function (baseline spirometry), atopy (specific IgE and general IgE), obesity (Body mass index (BMI)), age at diagnosis, and age and gender.

## 2. Materials and Methods

### 2.1. Design of the Study

In an observational study of the clinical significance of atopic status and pulmonary function in a cohort of 211 Bulgarian children with asthma, complex clinical instrumentation and laboratory studies were conducted following written informed consent. In the cluster analysis, only the children with complete data for Forced expiratory volume in 1 s (FEV1), BDR, age of asthma diagnosis, family history, BMI, total and specific IgE levels, ACQ score, and history of exacerbations were included.

The design of the study is presented in Figure 1.

### 2.2. Study Subjects

All children underwent clinical, instrumental, and laboratory studies. Basic epidemiological characteristics of children are presented in Table 1. Seventy-one of the enrolled children with asthma have complete data from all tests and analysis, and cluster analysis was applied.

We collected a detailed history of the onset and clinical course of asthma, comorbidity presence, and control treatment stage for the previous eight weeks before inclusion in the study. The level of asthma control was assessed with the ACQ questionnaire, validated for children 6 to 16 years old, and the interview-based version of the questionnaire—For those aged 6–10 years, has been performed by a trained interviewer. The 6 point scale of the questionnaire is used for cluster analysis, excluding the 7th question concerning the FEV1 spirometric index. The 1.50 threshold for “good control” is used.

Anthropometric indicators—height and weight—were removed for all children studied, and body mass index (BMI) was calculated.

All parents signed their informed consent for inclusion of their children. Additionally, all children older than 12 years signed an informed assent on their own before they participated in the study. The study was conducted following the Declaration of Helsinki, and the protocol was approved by the Ethics Committee of the Medical University of Sofia (Project identification code—no. 35D/2013, grant no. 23D/2013, Council of Medical Science).

### 2.3. Spirometry

Lung function testing was performed by conventional spirometry with a Masterscreen Pneumo spirometer ‘98 apparatus (Jager^®^, Wuerzburg, Germany) before and after administration of a short-acting bronchodilator (Salbutamol) according to the ERS/ATS quality and reproducibility criteria [15,16].

### 2.4. Immunological Methods

The assessment of atopic status was carried out by the serological examination of total IgE antibodies by ELISA (EUROIMMUN Total IgE ELISA, Medizinische Labordiagnostica AG). The kit is intended to screen for allergy diagnostics using an indirect sandwich ELISA. The microtiter plate is coated with polyclonal anti-human IgE antibodies. After incubation with 1:10 diluted patients‘ sera, incubations with POD-labelled anti-human serum and chromogen substrate solution followed. The optical density of the enzymatic reactions in the wells was measured photometrically at 450/630 nm, and a 4 point calibration curve was created to obtain the concentrations of total IgE in the samples.

The specific IgE antibodies against aero- and nutritional allergens were assessed by semi-quantitative blot immunoassay—Euroline Allergy Profile Pediatrics, Enzyme Allergo Sorbent Test (Enzyme Allergo Sorbent Test (EAST)) with Euroimmune^®^ (Medizinische Labordiagnostica, AG, 2014, Lϋbeck, Germany). The EUROLINE Pediatric (IgE) test kit determines semiquantitatively and in vitro specific IgE antibodies to 28 different inhalation and food allergens: gx (grass mix 2—Timothy grass, cultivated rye), t3 (birch), w6 (mugwort), d1 (der. Pteronyssinus), d2 (der. Farina), e1 (cat), e2 (dog), e3 (horse), m2 (Cladosporium her.), m3 (Aspergillus fum.), m6 (Alternaria alt.), f1 (egg white), f75 (egg yolk), f2 (cow’s milk), f3 (codfish), f76 (Lactalbumin), f77 (Lactoglobulin), f78 (casein), e204 (bovine serum albumin), f4 (wheat flour), f9 (rice), f14 {soybean), f13 (peanut), f17 (hazelnut), f31 (carrot), f35 (potato), f49 (apple), CCD (CCD marker). The used enzyme conjugate was alkaline phosphatase-labeled anti-human IgE (mouse), and the substrate solution (Nitroblue tetrazolium chloride/5-Bromo-4-chloro-3-indolylphosphate (NBT/BCIP). After stopping the reaction, the incubated test strips were placed on the adhesive foil of the green work protocol (created beforehand in the EUROLineScan program). Using the software program EUROLineScan, the intensity of the bands was calculated in Enzyme-Allergo-Sorbent Test (EAST) classes from 0 to 6.

### 2.5. Statistical Methods

Following the removal of missing data, a hierarchical cluster analysis was performed using Ward’s method in 71 (23 girls and 48 boys) of all 211 children included in the study, for whom there is a complete record of all data from the full set of studies performed. A hierarchical clustering method creates a pyramid or ‘hierarchical’ cluster of homogeneous clusters, which can be visually represented as a dendrogram (Figure 2).

We conducted a cluster analysis to identify subtypes of childhood asthma in the study population, based on key disease characteristics such as—Symptoms (control level, ACQ score), pulmonary function (baseline spirometry), atopy (specific IgE and general IgE), obesity (BMI), age at diagnosis, and patient characteristics—age and gender. The variables used in the analysis were defined after excluding multicollinear and clinically uninformative variables.

To determine whether patients in a cluster group are the same or different, Ward’s method was applied. The number of likely clusters was determined by visual inspection of the dendrogram (Figure 2). A k-means clustering method was also applied to define clusters definitively. The characteristics of the patients in the five clusters were compared with post-hoc analysis between groups (Bonferroni correction) and the chi-square method. Statistical analysis was performed with a software package for statistical analysis (SPSS^®^), IBM 2009, version 19 (2010), and Excel (v. 2010). Significance level α = 0.05 was chosen, i.e., for *p* < α, the null hypothesis is rejected.

## 3. Results

Using this cluster analysis with Ward’s method, the tested population of children was divided into five clusters that differed sufficiently in their cluster center and squared width (Figure 3 and Figure 4). The distribution of the specific IgE levels according to ACQ score in different cluster groups is presented at Figure 3. It is evident that two of the clusters have mainly poor asthma control and surprisingly, they are not the ones with the highest sIgE. As it is known, asthma in childhood is mainly allergic disease, but in our results, it is evident that children with the highest sIgE had relatively good control.

When we gathered all three aspects of atopy level, lung function and severity, we found that children with the most affected lung function were in those two clusters with poor control, but the ones with earlier diagnosis have more atopic features compared to the ones with later diagnosis (Figure 4).

A k-means cluster method was also applied to define clusters definitively. The characteristics of the patients in the five clusters were compared with the one-way ANOVA, and the chi-square method is presented in Table 2 and Figure 5.

Cluster characteristics are the following:

CLUSTER 1 (n = 10): Non-atopic, with good control.

The mean age of children in the cluster was 10.40 ± 3.32 years. Children in this group were characterized by the lowest levels of total IgE level, good control (low ACQ score 0.79 ± 0.24), and the median age of asthma diagnosis (78 ± 24 months which equals 6.5 ± 2 years). Children in this group had predominantly normal basal pulmonary function (baseline FEV1—95.24 ± 6.34% pred), with significant but not very high bronchial reversibility (BDR 17.65% ± 7.25).

CLUSTER 2 (n = 28): Mixed with poor control.

Children in this cluster were at a mean age of 10.3 ± 3.23 years with male prevalence, as in the general group of children included in the study and the cluster analysis. The cluster is characterized by high BMI (18.85 ± 0.79 kg/m^2^), the eldest age at asthma diagnosis (101 ± 53 months or 8.4 ± 4.4 years), poor asthma control (high ACQ score of 2.13 ± 0.32), significant bronchial reversibility (BDR)—19.77% ± 7.89) and mean IgE level (197.8 U/mL).

CLUSTER 3 (n = 19): Atopic with poor control and low lung function.

Children from this cluster were with low lung function and significant bronchial reversibility; children in this cluster were at a mean age of 12.9 ± 2.54 years. Characterized by early diagnosis of asthma (65 ± 14 months or 5.4 ± 1.2 years), decreased pulmonary function (low FEV1—82.28 ± 5.84%pred) and significant bronchial hyperreactivity (highest BDR 26.46 ± 13.4), with the highest worse clinical control (highest ACQ score, 2.32 ± 0.64), the highest titer of total IgE among clusters (486.4 U/mL) and the second highest result from specific IgE (0.269 ± 0.12 IU/mL). The cluster is dominated by the female gender, despite the general tendency to be dominated by the male gender, both in the general group of children and in the group of children included in the cluster analysis.

CLUSTER 4 (n = 9): Atopic, with partial clinical control, normal baseline pulmonary function, lowest bronchial lability.

Children in this cluster were at a mean age of 13.6 ± 2.96 years and mean age of diagnosis 84 ± 67 months (equaling 7 ± 5.6 years) with high titer of specific IgE (0.404 ± 0.27 IU/L), a high titer of total IgE (415.1 U/mL), relatively high BMI (18.42 ± 1.18 kg/m^2^) and partially controlled clinical control—Mean ACQ score (1.14 ± 0.18). Children in this cluster were characterized by normal baseline spirometry (mean FEV1 98.21 ± 4.88% pred) and significant but lowest bronchial hyperreactivity compared to the other four clusters (BDR 14.50 ± 5.67%).

CLUSTER 5 (n = 5) Non-atopic, very well controlled, the onset of asthma in preschool age.

Children in this group were the youngest—the mean age was 9.87 ± 3.72 years, with a significant female predominance, diagnosed with asthma in preschool age (48 ± 21 months or 4 ± 1.8 years). Children were low in body weight (lowest BMI of 14.36 ± 1.37), with excellent clinical asthma control (lowest ACQ score—0.50 ± 0.44). With regard to atopic status, they had a very low clinically insignificant titer of specific IgE (0.124 ± 0.07 IU/L) and average levels of total IgE (388.8 U/mL). Children had slightly decreased basal pulmonary function (FEV1—89.0 ± 7.52% pred) and high bronchial reactivity (large BDR—22.6 ± 8.54%).

Children with severe asthma, defined by The Global Initiative for Asthma (GINA) are abundant in Cluster 3 (65% of patients in this cluster). Only one child with severe asthma falls into another group—Cluster 2.

## 4. Discussion

“Asthma is not a separate disease, but a common concept for a number of different diseases, each of which is caused by a distinct underlying pathophysiological mechanism.” [17]. The Trousseau Asthma Program (TAP) cohort in France used a hierarchical cluster analysis by implementing Ward’s method [18,19]. In the study group of 551 preschool children, three clusters were identified: light episodic viral wheezers, atopic multi-trigger wheezers and non-atopic “poorly controlled” wheezers [20], which remain stable up to the age of 5 years [18]. However, at school age, the three clusters changed to: “asthma with severe exacerbations and multiple allergies,” “severe asthma with bronchial obstruction,” and “mild asthma” [19]. Fitzpatrick et al. Analyzed, with Ward’s hierarchical cluster analysis, the data for 161 children aged 6–17 years in the Severe Asthma Research Program (SARP) multicenter study and identified the following phenotypes: “symptomatic late-onset asthma,” “early-onset atopic asthma and normal pulmonary function,” “Early-onset atopic asthma and moderate comorbidity obstruction” and “early-onset atopic asthma with pronounced bronchial obstruction” [21].

We applied cluster analysis using Ward’s method, by which we defined five clusters in the examined group of children with asthma—two atopic, two non-atopic and one mixed. Children with severe asthma, defined by the GINA criteria, account for the highest percentage of atopic cluster 3 (65% of patients in this cluster), characterized by early onset of asthma (65 ± 14 months), high bronchial hyperreactivity (highest BDR), with the worst control (highest ACQ score) and the highest total and specific IgE titer. The expected cluster of girls, non-atopic overweight, or obesity was not defined in the group of children included in our analysis. One of the reasons for that is the lower percentage of girls in the general group and the high rate of atopy among women. We did not identify any girls with obesity and negative total and specific IgE antibodies.

Serum concentrations of total IgE are known to have high specificity but low sensitivity in determining atopic status. In the presence of elevated total IgE, the patient is likely to be atopic, but if normal, atopy cannot be ruled out [22]. The level of total IgE among atopic subjects correlates with the size of the target organ, with the lowest values reported in individuals with allergic rhinitis, highest in those with atopic dermatitis and intermediate for asthmatics [23]. Sensitization to allergens from various sources is often observed in children with persistent asthma, and the presence of allergies affects both the course of the disease and the manifestation of symptoms [24]. The literature suggests that multiallergen screening methods for the detection of aeroallergens in combination with a food allergen mix, such as that used in the present work, are more effective in characterizing the atopic status of children with asthma than measuring individual allergen-specific IgE [25].

IgE was valuable to determine the atopic groups, while we were surprised that the group with worse control didn’t show highest result for specific IgE. The highest one had the group with partial control. We looked in depth for any difference in the type of allergens showing higher specific IgE in both groups and we found a significantly higher number of children with poli-sensibilization (positive sIgE ≥ 0.35 IU/L or EAST GRADE 1 against three allergens) in the worse control group, which correlates with the highest total IgE level. In the general asthmatic group, we found a higher percentage of atopic children (positive sIgE ≥ 0.35 IU/L or EAST GRADE 1 against at least one allergen) in the poor control group. The poor asthma control correlates with the presence of aeroallergen sensibilization in general and with the sensibilization against D. farinae (titer ≥ 1 EAST class, *p* = 0.006) and against D. pteronyssimus (only at a higher titer ≥ 3 EAST class, *p* = 0.004) in particular. The results of several studies using cluster analysis support the hypothesis that obesity is associated with two asthma phenotypes—a non-atopic late-onset phenotype that can be associated/”triggered” by obesity and an early-onset asthma phenotype in childhood age, usually atopic, which may be aggravated as a clinical course by obesity [26,27,28,29,30].

The claim that obesity affects asthma control and exacerbations remains controversial and speculative in the literature. There is also no strong evidence for an association between BMI and loss of pulmonary function in children with asthma, beyond the usual effects of obesity on lung volumes [29]. In the study group of children with asthma, in more than two-thirds of children, we found them to be underweight or at normal weight. Nineteen percent of children weigh above the upper limit of normal, and 10% presented with obesity. There was a statistically insignificant difference in the mean BMI of children with asthma and healthy controls (18.87 ± 3.00 vs. 17.34 ± 3.90). As expected, we have not demonstrated a significant relationship between BMI and asthma control, baseline spirometry, BDR, atopic status (own atopic terrain, general and specific IgE antibodies levels), total for the asthma population studied, and age and gender sampling. The only relationship we found between BMI and pulmonary function was that children with baseline FEV1 < 60% had significantly higher BMI compared to children with mild bronchial obstruction. A weak but significant correlation of BMI with BDR, calculated as the absolute change (in % and in mL), was found to be associated with the indicator of the age, height and weight of the children.

## 5. Conclusions

This study is the first cluster analysis of a Bulgarian population of pediatric asthma patients. We defined five separate clusters with major determinants of age at diagnosis of asthma, baseline spirometry, bronchial reactivity, BMI, clinical control and atopic status, defined by the levels of general and specific IgE antibodies.

In the study group, more than two-thirds of the asthmatic children were normal or underweight. Contrary to expectations, a cluster of girls, non-atopic overweight or obese, was not observed within the clusters formed within our results. Children with severe asthma, according to GINA, account for the highest percentage of atopic Cluster 3 (65% of patients in this Cluster), characterized by early onset of asthma (65 ± 14 months), high bronchial reversibility (the highest BDR), with the worst control (highest ACQ score) and highest total IgE titer.

The application of new statistical methods based on machine learning allows for better differentiation of groups of children with a similar course of the disease. This makes it possible to individualize the therapeutic drug and non-drug plan for treatment and follow-up of these children, and, as a final result, to achieve and maintain better clinical control.

## Figures and Tables

**Figure 1 antibodies-09-00031-f001:**
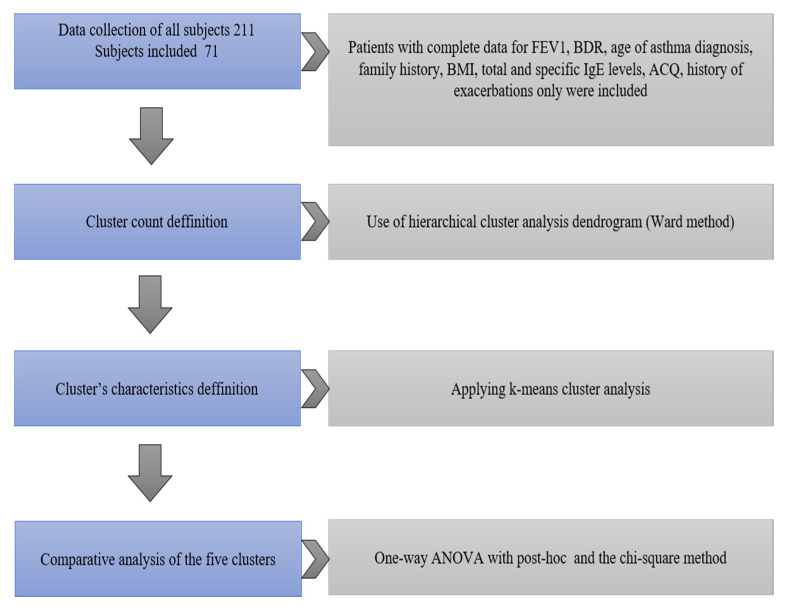
Design of the study. Forced expiratory volume in 1 s (FEV1), Bonchodilator response (BDR), Body mass index (BMI), Asthma Control Questionnaire (ACQ).

**Figure 2 antibodies-09-00031-f002:**
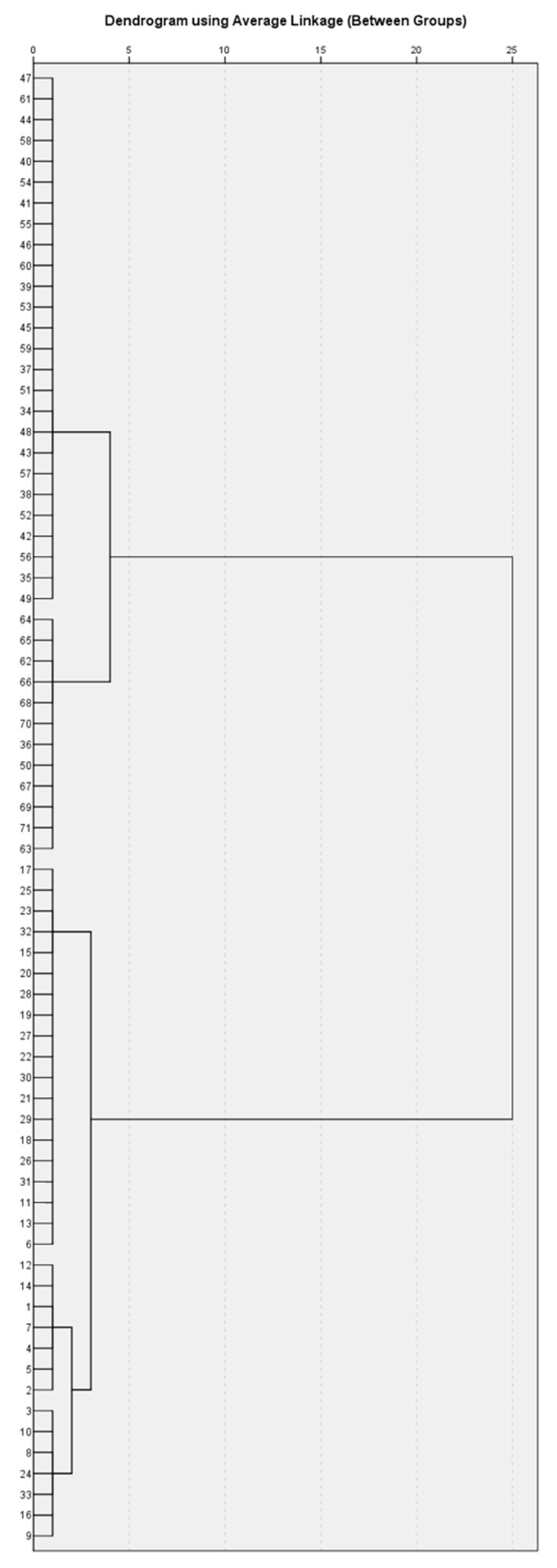
Dendrogram obtained using Ward’s method in the studied 71 patients.

**Figure 3 antibodies-09-00031-f003:**
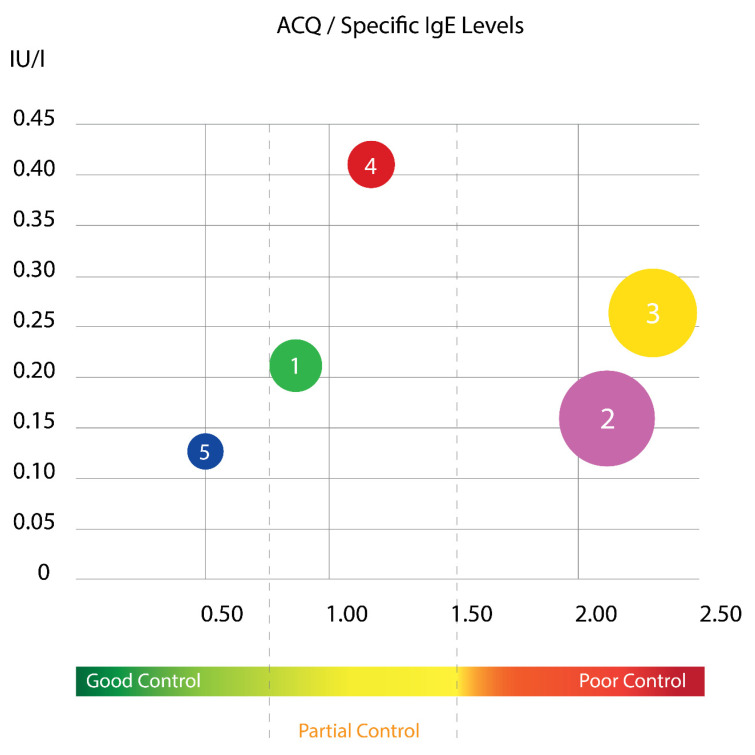
Representation of the five clusters according to mean levels of specific IgE and ACQ Score. Each color represents different cluster (cluster 1, green; cluster 2, purple; cluster 3, yellow; cluster 4, red; cluster 5, dark blue). The circled area represents the size of the population distributed in each cluster. (Left Side of the axis X before the first dotted line (0–0.75)—good control; between two dotted lines (0.75–1.5)—partial control; right side after the second dotted line (over 1.5)—bad control).

**Figure 4 antibodies-09-00031-f004:**
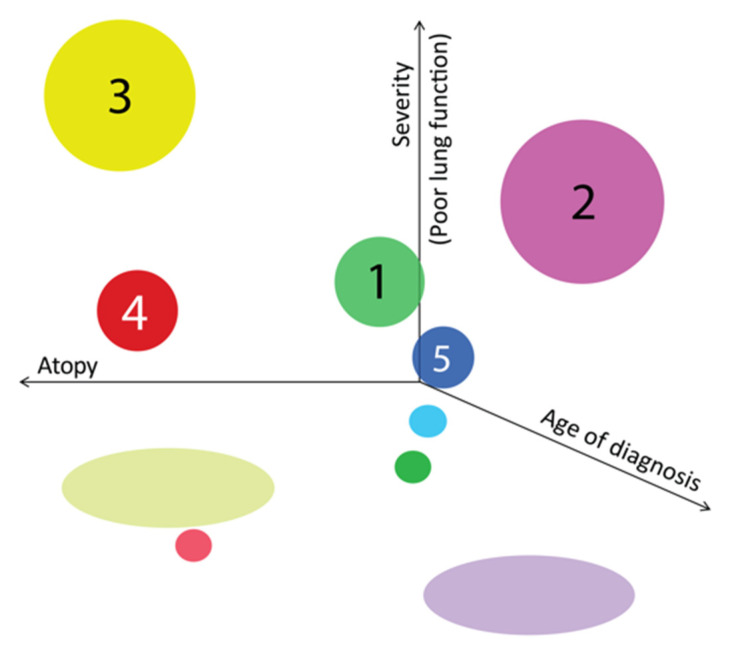
3D representation of the five clusters according to atopy (X axis—on the left the atopic features increase), age of diagnosis (Z axis—on the right the age increases) and lung function results (Y axis—upwards the FEV1 decreases) Each color represents a different cluster (cluster 1, light green; cluster 2, light purple; cluster 3, yellow; cluster 4, red; cluster 5, dark blue). Sphere size represents the size of the population distributed in each cluster.

**Figure 5 antibodies-09-00031-f005:**
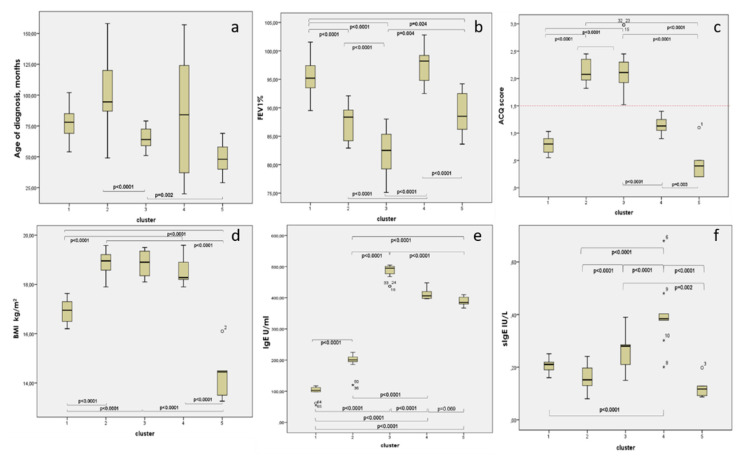
Box-plots (with mean and SD) for each characteristic in clusters and all of the significant differences (on post-hoc analyses between clusters) noted: (**a**) age of diagnosis; (**b**) FEV1%; (**c**) ACQ score; (**d**) BMI; (**e**) IgE; (**f**) sIgE. Dotted line on ACQ represents the cut off for good control. (ANOVA, post-hoc, Bonferroni correction).

**Table 1 antibodies-09-00031-t001:** The demographic structure of individual groups of respondents. Data are presented as a number (%) or mean ± SE.

Subject Group	Number	Mean Age (Range), Years	Boys	Girls
Main group asthmatic children	211	10.2 (4–17.6)	134	77
Asthmatic children in the cluster analysis group	71	10.9 (5.3–17.7)	34	37
Healthy controls	46	10.92 (6–16.4)	18	28

**Table 2 antibodies-09-00031-t002:** Cluster characteristics of the study subjects.

*Characteristics*	Cluster 1	Cluster 2	Cluster 3	Cluster 4	Cluster 5	Level of Significance, *p*
	Non-Atopic, Good Control	Mixed, Bad Control	Atopic, Bad Control	Atopic, with Partial Clinical Control	Non-Atopic, very Well Controlled	
*Subjects, n*	10	28	19	9	5	
*Boys (%)*	40%	60.71%	42.10%	55.5%	20%	0.005
*Age, years (mean ± SD)*	10.40 ± 3.32	10.3 ± 3.23	12.9 ± 2.54	13.6 ± 2.96	9.87 ± 3.72	0.079
*Age of diagnosis, months (mean ± SD)*	78 ± 24	101 ± 53	65 ± 14	84 ± 67	48 ± 21	0.0001
*ACQ score (mean ± SD)*	0.79 ± 0.24	2.13 ± 0.32	2.32 ± 0.64	1.14 ± 0.18	0.50 ± 0.44	0.004
*BMI, kg/m^2^ (mean ± SD)*	16.91 ± 0.72	18.85 ± 0.79	18.09 ± 2.5	18.42 ± 1.18	14.36 ± 1.37	0.27
*FEV_1_%pred (mean ± SD)*	95.24 ± 6.34	87.50 ± 4.62	82.28 ± 5.84	98.21 ± 4.88	89.0 ± 7.52	0.002
*BDR ∆FEV1%* *(mean ± SD)*	17.65 ± 7.25	19.77 ± 7.89	26.46 ± 13.4	14.50 ± 5.67	22.6 ± 8.54	0.0006
*ELISA total IgE (U/mL)*	97.0	197.8	486.4	415.1	388.8	0.0003
*Elevated specific IgE (IU/L)*	0.209 ± 0.042	0.160 ± 0.081	0.269 ± 0.12	0.404 ± 0.27	0.124 ± 0.07	0.009

Abbreviations: ACQ (Asthma Control Questionnaire), FEV1 (Forced expiratory volume), BDR (Bronchodilator response).

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
