# Peer review of "Identification of Specific IgE Antibodies and Asthma Control Interaction and Association Using Cluster Analysis in a Bulgarian Asthmatic Children Cohort"

_2073-4468, 2020, doi:10.3390/antib9030031_

Round 1
Reviewer 1 Report
The authors present a study on asthma-related phenotype identification, having analyzed a group of 71 children from Bulgaria. For this purpose, they evaluated pulmonary function, IgE and some allergens specific IgE titers. They also have considered the results of a questionnaire, the Body Mass Index, age and sex of the children.
The work is interesting but from my point of view it does not fit in a journal on antibodies. It should be noted that in this study an immunochemical kit was used to semi-quantify specific IgE for some allergens, which are never mentioned in the article. The title of IgE is also determined, but in materials and methods it is not clear what method was used.
In my opinion, there is no broad discussion on the results of the immunochemical assays for the different clusters, in particular for 3 and 4.
In addition, there are other issues that were not clear to me:
- what are the results of the eosinophil counts mentioned in the abstract?
- the gender characterization does not seem consistent throughout the text (data from table 1, rows 125, 194-195, 206-207)
- have the clusters defined "definitively" in the dendogram been redefined according to the QA and the title in IgE? Figure 3 lacks the identification of the axes in the graph and the units of the ordinates.
- Why weren't exactly the same colours used in the Illustrations 3 and 4?
- the legend of figure 4 isn`t clear. In the graph where is the dotted line?
- table 2: how was "elevated specific IgE” calculated? Units?
- the graphs of figure 5 are not clearly seen
- line 188: lowest ACQ? Isn't cluster 5 lower?
- line 218:preschool?
-lines 247-248: the highest specific IgE titer seems to be for cluster 4, not for 3
- line 268: what is BDI? There are acronyms that appear without identification and for some the identification only appears later in the text.
Author Response
Dear Editor,
Dear reviewers,
Thank you for your time to revise our Manuscript ID: antibodies-799643, entitled “Identification the specific IgE antibodies and asthma control interaction and association using cluster analysis in a Bulgarian asthmatic children cohort". We acknowledge that our paper might have some issues in conformity with the following comments.
Reviewer 1
Open Review
(x) I would not like to sign my review report
( ) I would like to sign my review report
English language and style
(x) Extensive editing of English language and style required
( ) Moderate English changes required
( ) English language and style are fine/minor spell check required
( ) I don't feel qualified to judge about the English language and style
|
Ø Thank you for your valuable comments and the good overall evaluation of the manuscript. We agree with the referee that we are not native English speakers. However, we checked the English once again with the premium version of Grammarly, and the final version was approved by a colleague of ours who is a native English speaker.
|
Yes |
Can be improved |
Must be improved |
Not applicable |
|
Does the introduction provide sufficient background and include all relevant references? |
( ) |
(x) |
( ) |
( ) |
|
Is the research design appropriate? |
( ) |
( ) |
(x) |
( ) |
|
Are the methods adequately described? |
( ) |
( ) |
(x) |
( ) |
|
Are the results clearly presented? |
( ) |
( ) |
(x) |
( ) |
|
Are the conclusions supported by the results? |
( ) |
( ) |
(x) |
( ) |
Comments and Suggestions for Authors
The authors present a study on asthma-related phenotype identification, having analyzed a group of 71 children from Bulgaria. For this purpose, they evaluated pulmonary function, IgE and some allergens specific IgE titers. They also have considered the results of a questionnaire, the Body Mass Index, age and sex of the children.
The work is interesting but from my point of view it does not fit in a journal on antibodies. It should be noted that in this study an immunochemical kit was used to semi-quantify specific IgE for some allergens, which are never mentioned in the article. The title of IgE is also determined, but in materials and methods it is not clear what method was used.
- Thank you for the note. Indeed, оur research question was devoted to Bulgarian pediatric patients with asthma and their assessment of pulmonary function, IgE, and some allergens specific IgE titers.
- We agree partially with your assertion that our paper does not refer to the main topics mentioned in the Aim and Scope of Antibodies. Still, we do use total and specific IgE antibodies to define our clusters. Moreover, we have been invited to submit a paper, and our title was approved by the managing editor as following the aim and the scope at first glance. We believe that if we address all the issues pointed out by the referees, our paper will be considered as appropriate for the journal.
- We strongly agree with the referee that the method for specific IgE used was semi-quantity analysis – immunoblot (blot immunoassay). We`ve added the missing information. The total IgE was assessed by ELISA, as mentioned in the Material and methods, Immunological methods section.
In my opinion, there is no broad discussion on the results of the immunochemical assays for the different clusters, in particular for 3 and 4.
- Thank you for the constructive critic. We have extended the discussion on these clusters and added references (22-25), but it was a bit challenging since there are not enough relevant references in the literature, especially for children.
In addition, there are other issues that were not clear to me:
- what are the results of the eosinophil counts mentioned in the abstract?
- Thank you for the note. We have the data for eosinophil counts in nasal smears. Still, we found that most of the authors report a very high number of eosinophils in children with allergic rhinitis without local corticosteroid therapy and a very low number in atopic children with allergic rhinitis on local therapy. For proper evaluation, we should have analyzed only the children without local therapy, and thus we would have missed more than half of our patients. To be more practically oriented (as most of the children with allergic rhinitis worldwide receive local therapy, our cluster analysis didn’t include the eosinophilic data. Therefore we corrected the abstract with omitting the eosinophil count.
- the gender characterization does not seem consistent throughout the text (data from table 1, rows 125, 194-195, 206-207)
- The Referee assertion is right. We have double checked our data, and we found there was a technical error in the total number of boys and girls, respectively, their numbers were inverted. We have corrected the issue. This did not affect the cluster analysis since these total numbers were not included in the analysis.
- Regarding the asthmatic children in the cluster analysis group, there was a miscalculation in the count. There were 34 boys and 37 girls, instead of 33 and 38, respectively. We have revised the first table accordingly.
- have the clusters defined "definitively" in the dendogram been redefined according to the QA and the title in IgE? Figure 3 lacks the identification of the axes in the graph and the units of the ordinates.
- Thank you for the valuable comment. We have fixed the figure 3 by adding the axes. We agree it was not clear before. Now, we believe that it is more understandable. The dendrogram was based on the complete analysis, including also the IgE levels, ACQ, etc.
- Why weren't exactly the same colours used in the Illustrations 3 and 4?
- We support the referee's assertion that the colors are not the same. We have asked a professional help from а statistician and graphic designer for correcting the mismatch of the colors to be more clear and to follow the color code.
- the legend of figure 4 isn't clear. In the graph where is the dotted line?
- We agree with the referee that due to the poor quality of the previous figure, the dotted line was not visible. This issue is corrected accordingly.
- table 2: how was "elevated specific IgE" calculated? Units?
- Thank you for the valuable note. Indeed, we did not mention the units. We have added this missing information where it is applicable.
- the graphs of figure 5 are not clearly seen
- Thank you for the note, and we apologize for the inconvenience. We provided a figure with a higher resolution during the submission, and if needed, we will provide another one with better quality. The figure in the text might be distorted because of the word file during submission.
- line 188: lowest ACQ? Isn't cluster 5 lower?
- We agree with the referee that the lowest value is in cluster 5. It was a technical error. The ACQ was low but not the lowest. The mistake was corrected in the text.
- line 218:preschool?
- Thank you for the valuable note. In Bulgaria, the children start school at the age of 7 years. Cluster 5 had asthma diagnosed by the age of 4, which is preschool. We agree there are some countries where children start school earlier. However, we checked that according to the universal definition, four-year-old children are in the preschool age. We did not notice that in the text, we have also written infancy (despite 4 years is late for infancy). Therefore we corrected the term in both places as preschool age.
-lines 247-248: the highest specific IgE titer seems to be for cluster 4, not for 3
- Thank you for the valuable note. We did not notice that technical mistake, and we put a couple of sentences in the discussion regarding this finding.
- line 268: what is BDI? There are acronyms that appear without identification and for some the identification only appears later in the text.
- Thank you for the note. It seems that we misused this acronym. We intended to mean BDR and not BDI. We have revised the text accordingly.

Reviewer 2 Report
In this present study investigated “Identification the specific IgE antibodies and asthma control interaction and association using cluster analysis in a Bulgarian asthmatic children cohort”. There are some concerns to be addressed:
1.Whether boy or girl effected the result in Cluster 1-5.
2.table 2 “Cluster Non-atopic good control “ should modify as Cluster 1 Non-atopic good control.
3.Figure 5 the result also were showed in table 2. I suggest to delete
4.How to distinguish between specific IgE and general IgE?
5.Conclusions need to be more streamlined.
.
Author Response
Dear Editor,
Dear reviewers,
Thank you for your time to revise our Manuscript ID: antibodies-799643, entitled “Identification the specific IgE antibodies and asthma control interaction and association using cluster analysis in a Bulgarian asthmatic children cohort". We acknowledge that our paper might have some issues in conformity with the following comments.
Reviewer 2
Open Review
(x) I would not like to sign my review report
( ) I would like to sign my review report
English language and style
( ) Extensive editing of English language and style required
( ) Moderate English changes required
(x) English language and style are fine/minor spell check required
( ) I don't feel qualified to judge about the English language and style
Yes Can be improved Must be improved Not applicable
Does the introduction provide sufficient background and include all relevant references?
( ) (x) ( ) ( )
Is the research design appropriate?
( ) (x) ( ) ( )
Are the methods adequately described?
( ) (x) ( ) ( )
Are the results clearly presented?
( ) (x) ( ) ( )
Are the conclusions supported by the results?
( ) (x) ( ) ( )
Comments and Suggestions for Authors
In this present study investigated "Identification the specific IgE antibodies and asthma control interaction and association using cluster analysis in a Bulgarian asthmatic children cohort". There are some concerns to be addressed:
- Thank you for your valuable comments and the overall positive evaluation of the manuscript.
1.Whether boy or girl effected the result in Cluster 1-5.
- Thank you for the comment, we agree that it is not very clear, but the answer to the questions is both genders. We have put only the boys' percentage in the table for shortening the rows, believing that the rest percentage would be easily calculated if someone needs the number of girls.
2.table 2 "Cluster Non-atopic good control "should modify as Cluster 1 Non-atopic good control.
- Thank you for the valuable note, we corrected the issue with the variant proposed by the referee.
3.Figure 5 the result also were showed in table 2. I suggest to delete
- Thank you for the suggestion. The referee is right to point out that there is some similarity between the respective figure and table. However, we provide both for these reasons – the table contains the numbers, which can be very useful for other investigators and authors, including for systematic reviews and meta-analysis. At the same time, the figure is very representative, and for most readers, it will be more beneficial to figure out the data. Therefore, we would be delighted if we can keep both in the paper.
4.How to distinguish between specific IgE and general IgE?
- Thank you for the note. We acknowledge that the total IgE levels are characterized by high specificity but low sensitivity. Some atopic children could be misdiagnosed as non-atopic by using only total IgE. Additionally, the specific IgE were tested with a panel of the most common allergens (in our study, 11 aeroallergens, and 15 food allergens); therefore, if a child is allergic to something not included in the kit would also be misdiagnosed. To eluding such possibilities, the best approach is to use both tests. For this reason, we combined the methods and put them separately in the cluster analysis.
- Conclusions need to be more streamlined.
- Thank you for the constructive critic. We agree with the referee that the conclusions might have been a bit vague. However, this is the first cluster analysis in asthma in Bulgarian children; thus, we tried to estimate the power of our results and draw some initial conclusions. Probably a bigger cohort in the future could give us more concise answers, especially for some of our clusters where the number of patients is relatively small. We gave our best to improve our conclusion based on our results.

Round 2
Reviewer 1 Report
Having analysed the authors' replies and the changes made to the document, I consider that the quality has increased considerably. However, I am sorry but I still have doubts about the immunochemical assays. In the materials and methods the authors keep only names of immunochemical kits, which I cannot find clear information about and do not make any description of them. For example, for dosage of "common?" IgE was performed ELISA. Indirect? Which Ags and secondary Ab were used? It's unclear to me. Therefore, I consider that the article should not be published in this journal.
Author Response
Having analysed the authors' replies and the changes made to the document, I consider that the quality has increased considerably.
- Thank you for your time to revise our paper again and for the overall evaluation of the revised manuscript.
However, I am sorry but I still have doubts about the immunochemical assays. In the materials and methods the authors keep only names of immunochemical kits, which I cannot find clear information about and do not make any description of them. For example, for dosage of "common?" IgE was performed ELISA. Indirect? Which Ags and secondary Ab were used?
- We acknowledge that this information is not enough. The reason for that is the fact that we used commercially available reagents and kits. Nevertheless, we would like to address this issue, and for that reason, we have added the following (can be seen via track changes):
- The word “common” was replaced by “total.” This was a technical mistake during writing the first draft.
- We have added the details about the ELISA for the determination of total IgE, including the type of ELISA and principle of the method, the reagent antibodies used, and how the total levels of IgE were calculated in the patients` samples (lines 124-129).
- Regarding the semiquantitative blot analysis for determination of specific IgE antibodies – we have added all the required information, including the inhalation and food allergens included in the kit, the secondary conjugated antibody, the substrate solution, and the calculation of the results based on the automated scanning and software use (lines 133-145).
It's unclear to me. Therefore, I consider that the article should not be published in this journal.
- We are sorry to hear that the information in the Material and Methods section is insufficient. However, after adding the missing information, we hope that this invited paper would be suitable for publication in the journal.

Reviewer 2 Report
accept
Author Response
- Thank you for your time to revise our paper again and for the overall evaluation of the revised manuscript.